# *Ocimum campechianum* Mill. from Amazonian Ecuador: Chemical Composition and Biological Activities of Extracts and Their Main Constituents (Eugenol and Rosmarinic Acid)

**DOI:** 10.3390/molecules26010084

**Published:** 2020-12-27

**Authors:** Massimo Tacchini, Monica Paulina Echeverria Guevara, Alessandro Grandini, Immacolata Maresca, Matteo Radice, Letizia Angiolella, Alessandra Guerrini

**Affiliations:** 1Pharmaceutical Biology Laboratory, Technopole Terra&Acqua Tech (Research Unit 7), Department of Life Sciences and Biotechnology, University of Ferrara, P.le Luciano Chiappini 3, Malborghetto di Boara, 44123 Ferrara, Italy; massimo.tacchini@unife.it (M.T.); alessandro.grandini@unife.it (A.G.); mci@unife.it (I.M.); 2Department of Earth Science, Universidad Estatal Amazónica, Puyo 160106, Ecuador; mecheverria@uea.edu.ec (M.P.E.G.); mradice@uea.edu.ec (M.R.); 3Department of Public Health and Infectious Diseases, Sapienza University of Rome, P.le Aldo Moro 5, 00165 Rome, Italy; letizia.angiolella@uniroma.it

**Keywords:** *Ocimum campechianum*, antioxidant activity, plant protection, cytotoxicity, synergistic activity

## Abstract

The essential oil (EO), the methanolic (MeOH), and the 70% ethanolic (70% EtOH) extracts obtained from the aerial parts of *Ocimum campechianum* Mill. (Ecuador) were chemically characterized through gas-chromatography coupled to mass spectrometry detector (GC-MS), high-performance liquid chromatography coupled to diode array-mass spectrometry detectors (HPLC-DAD-MS) and studied for their in vitro biological activity. The radical scavenger activity, performed by spectrophotometric 1,1-diphenyl-2-picrylhydrazyl (DPPH) and 2,2′-azino-bis(3-ethylbenzothiazoline-6-sulfonic acid) (ABTS) assays, highlighted significant IC_50_ values for the EO, extracts and their main constituents (eugenol and rosmarinic acid). EO (and eugenol) showed noteworthy activity against *Pseudomonas syringae* pv. *syringae* and a moderate effect against clinical *Candida* strains, with possible synergism in association to fluconazole against the latter microorganisms. The extracts and pure molecules exhibited weak cytotoxic activity against the HaCat cell line and no mutagenicity against *Salmonella typhimurium* TA98 and TA100 strains, giving indication of safety. Instead, EO showed a weak activity against adenocarcinomic human alveolar basal epithelial cells (A549). The above-mentioned evidence leads us to suggest a potential use of the crude drug, extracts, and EO in cosmetic formulation and food supplements as antioxidant agents. In addition, EO may also have a possible application in plant protection and anti-*Candida* formulations.

## 1. Introduction

Ecuador is well known for its biodiversity and is considered among the 17 megadiverse countries, counting about 10% of all plant species in the world and having the third highest density of endemic ones: about 4000 out of a total of 20,000 [1,2]. The aromatic plants of Amazonian Ecuador represent an appropriate renewable source for the production of essential oils and flavors, and an interesting economic alternative to sustainable development, with the perspective of generating wealth for this region. Numerous studies have been conducted in the Amazonian area of Ecuador, particularly describing the chemodiversity of essential oils as well as other natural bioactive molecules that may find application in food, beverages, cosmetics, pharmaceuticals, and pesticides [3,4,5,6]. However, about 50% of all the scientific publications on the botanical native species are related to only eight families (Arecaceae, Poaceae, Asteraceae, Fabaceae, Cucurbitaceae, Solanaceae, Orchidaceae, and Euphorbiaceae), while the remaining 246 are scarcely studied, with few or no publications on either phytochemistry or ethnopharmacology. As a result, many species are still little explored [7]. To complete this scenario, it should be noted that the overharvesting of crude drugs in the wild for the formulations of pharmaceutical and herbal products has often led to the depletion of valuable natural resources, and any program that attempts to promote their use must incorporate strategies for the sustainable sourcing of raw materials. The relationship between tropical biodiversity, conservation, and human health is complex and should not be oversimplified. The most effective way in which health and conservation can be combined to serve the needs of local and international communities is by incorporating this complexity into a package of complementary activities including the development of natural products [8,9]. Within this context, the cooperation among University of Ferrara (Italy), Amazonian State University (Ecuador), and other Ecuadorian Universities was born to perform, over the years from 1996 to now, various research activities on endemic species of the Amazon region, studying in particular essential oils of well-known genera of traditional medicine, such as *Piper*, *Ocotea*, *Citrus*, and *Croton*, and almost unexplored ones such as *Hedyosmum* and *Myrcia* [1,4,6,10,11,12,13,14], promoting the training of local researchers in Italy, through a PhD program, collaborating in the development of sustainable Ecuadorian supply chains.

In this research work we have dealt with *Ocimum campechianum* Mill. (synonym *Ocimum micranthum* Willd.), belonging to Lamiaceae family, an interesting native species of the South and Central American tropics, known as “Albahaca de campo” or “Albahaca silvestre” and widely used by indigenous population both for culinary and medicinal purposes. This species has already been studied in previous research works especially for leaf and aerial parts EO which has been shown to have antifungal, insect repellent and analgesic activities [10,15,16,17,18]. Only recently Ruiz-Vargas et al. (2020) [19] reported the isolation, identification, and biological evaluation of some secondary metabolites present in the leaf infusion of *O. campechianum*.

The aim of our research was the chemical characterization of the EO and, for the first time, of the hydroalcoholic extracts of aerial parts, combined with the evaluation of their biological activity and that of their main constituents, providing a more exhaustive approach for studying the health potential of this species. In fact, the combination of critically assessed compositional data and biological effects may increase the scientific soundness of research [20]. In addition, we tested in vitro preliminary phytopathogenic activity with the precise aim of identifying promising extracts for further investigations with respect to the phytoiatric application, or for the sustainable defense of the crops. 

## 2. Results and Discussion

### 2.1. Chemical Characterization

The EO content and composition were similar to our previous published results [10,18]. Briefly, the extraction yield (EO content) was 0.71 ± 0.01% and thirty-one compounds were identified corresponding to 97.1% of the total. The most abundant component (Table 1) was the monoterpene eugenol (43.6%), followed by 1,8-cineole (4.4%); among the sesquiterpenes, we detected β-caryophyllene (10.8%), β-elemene (8.1%), δ-elemene (4.2%), and bicyclogermacrene (2.9%), which is consistent with results reported by other authors [10,18].

Other chemotypes present in literature are characterized by methyleugenol (62.0%), recognized in the specimens collected from Pará State (Brazil) and Chocò (Colombia); by 1,8-cineole (61.8%) and sabinene (16.4%), identified in two specimens collected in Piauí State (Brazil); and by β-caryophyllene (78.6%) and 1,8-cineole (13.0%), which are the main compounds of the Maranhão (Brazil) and Germany *O. campechianum* EO [15,16,17,22].

Table 2 showed the composition of alcoholic extracts, chemically characterized for the first time in this research. The mass fragmentations of molecules were in according to previous literature [23,24]. The methanolic and hydro-alcoholic extracts with a yield of 5.3 ± 0.2% and 19.2 ± 0.2%, respectively, showed a similar fingerprinting where rosmarinic acid was the main compound, followed by rutin, caftaric and chlorogenic acids as minor components.

Previous research found in literature reported rosmarinic acid levels in *Ocimum basilicum* L. cultivars, varied from 0.06 mg/g to 12.7 mg/g on the dried matter, with caftaric acid levels ranging from 0.002 mg/g to 0.49 mg/g [25]. Our results highlighted an interesting composition of *O. campechianum* that could be suggested as an alternative for *O. basilicum*, with reference to the content of rosmarinic acid.

Ruiz-Vargas et al. (2020) reported the isolation from *O. campechianum* infusion of 5-demethyl nobiletin, 5-demethyl sinensetin, luteolin, methyl rosmarinate, and rosmarinic acid. Except the last compound the others have not been detected in our research [19].

### 2.2. Antioxidant Activity

All extracts exhibited a noteworthy antioxidant capacity in both performed tests (DPPH and ABTS). In particular, EO showed the highest radical scavenging activity in both assays (Table 3) with an IC_50_ of 7.7 ± 0.1 µg/mL against DPPH and 3.18 ± 0.29 µg/mL against ABTS, both close to the value obtained with the positive control (Trolox). This direct action of EO against the propagation of the prooxidative radical reaction could be due to the abundant presence of eugenol, although it should not be forgotten that the total antioxidant capacity of EO could also be the result of the complex interaction between the components in which synergistic or antagonistic actions intervene. In effect, the *O. campechianum* EO, even though it consists of about 44% eugenol, is also characterized by a good amount of cyclohexadiene-like components (~12% β-elemene and δ-elemene) that could work synergistically to offer high protection against oxidative processes [26]. The experimental results of the eugenol antiradical scavenger tests are further proof of the above, in fact they showed IC_50_ lower than EO and comparable to the positive control (Table 3), confirming previously data of literature [27]. Due to its antioxidant and anti-inflammatory effects, eugenol has been already studied for the treatment of irritant contact dermatitis [28], and also as components of active packaging to reduce the microbial decay and to preserve the antioxidant characteristics of table grapes [29], strawberries [30], and bayberries [31].

As for the other extracts, the hydroalcoholic one exhibited a bioactivity close to the one of EO, while the methanolic displayed a weaker activity when compared to the previous. Both ethanolic and methanolic extract are rich in rosmarinic acid, showing IC_50_ values close to Trolox, confirming previously data of literature [32], but the ethanolic one showed a slightly higher content of the other identified phenolic acids and flavonoids (Table 2) than the methanolic extract. Probably, the reasons for the modest result of radical scavenger activity of the latter phytocomplex could also be attributed to this factor. However, the results confirmed very promising antioxidant properties for all extracts. The positive evaluation of the direct antioxidant activity through two techniques (DPPH and ABTS tests), the considerable presence of rosmarinic acid and eugenol (well known in the literature for their health potential), and the studies already underway to exploit the high antioxidant potential of the latter two molecules suggest that these results can be considered reliable to predict good future prospects for the use of *O. campechianum*, crude drug and extracts, in the health and wellness market. 

### 2.3. Antimicrobial Activity of O. campechianum 

The assay was performed on a 96-well plate in microplate reader.

The tests performed against the Gram-positive bacteria considered (*Staphylococcus aureus,* ATCC 6538) did not show any noteworthy activities for the extracts and rosmarinic acid, while eugenol, already well known in the literature for its antibacterial capacity [27], exhibited a MIC value under the 2000 μg/mL (Table 4). Regarding the Gram-negative bacteria, instead, *P. syringae* pv. *syringae* is the most polyphagous bacterium in the *P. syringae* complex that primarily affects woody and herbaceous host plants. The data obtained revealed noteworthy phytopatogenic activity of *O. campechianum* EO, and eugenol, against *P. syringae* pv. *syringae*, pointing out this extract as an interesting candidate for future deeper investigation for phytoiatric application. Eugenol is an active substance yet approved by European Parliament and Council in 2013 for the formulation of plant protection products [33]. On the strength of our results, the EO of *O. campechianum* could also become a promising active substance for plant protection as potential succedaneum of clove oil, already included in European Regulation 540/2011 [34] and with eugenol as main compound. These considerations may be of interest for both the development of a local economy and the international trade of this species. Moreover, previous literature was not reach of antimicrobial studies about *O. campechianum* extracts, and the few founded highlighted the antimicrobial activity of *O. campechianum* EO using different methods [18] or adopting a method that is not clearly decipherable and reproducible [35]. Therefore, these data could enrich the panorama of the antimicrobial activity of *O. campechianum* extracts.

Starting from literature data regarding the effect of *O. campechianum* EO against *Candida* spp., we investigated its potential alone and in combination with fluconazole against *Candida* clinical strains [18,35]. Due to the increasing levels of *Candida* spp. resistance to antifungal agents, the research of other remedies, more effective and safer than the current ones, is of great interest [36]. The synergistic effect of eugenol in combination with fluconazole was already described in literature [26]. Taking this fact into account and considering the composition of *O. campechianum* EO, this crude extract could be a new potential remedy against *Candida* spp. resistance.

The MIC and MFC results that gave the best FIC_index_ data are reported in Table 5. The MIC and MFC values for EO against all three strains were moderate. Even if no synergistic effects were evidenced for all strains, the FIC_index_ for *C. glabrata* (FLU 43976) was close to the value of synergism, with the EO MICs and MFCs 2-fold lower than those displayed by the alone EO and for fluconazole 32-fold lower than those of alone fluconazole. This result suggested that *O. campechianum* EO could be an interesting candidate to deeply investigate the synergism against fluconazole-resistant *Candida* spp.

### 2.4. Cytotoxic Activity of O. campechianum 

EO, MeOH and 70% EtOH extracts of *O. campechianum* were subjected to the 3-(4,5-dimethylthiazol-2-yl)-2,5-diphenyl-*2H*-tetrazolium bromide (MTT) assay [37] for the evaluation of their cytotoxic effects on lung carcinoma (A549) during 24 h of exposure (Figure 1). The obtained data were compared with the values of the medium with dimethyl sulfoxide (DMSO) 0.1% (negative control), and doxorubicin was used as positive control (concentration range 0.1 to 20 μg/mL, IC_50_ = 2.92 ± 0.12 μg/mL).

A preliminary screening of the extracts revealed a negative outcome for the alcoholic ones, and a weak activity for the EO that exhibited an IC_50_ value of 71.89 ± 3.84 µg/mL. Rosmarinic acid was tested without obtaining any noteworthy results and eugenol did not reach an IC_50_ value at the highest concentration tested. As a matter of fact, Fangjun and Zhijia (2018) [38] showed that the latter molecule needs a concentration of at least 400 µM to obtain a result after 24 h of exposure. Therefore, the activity of the EO could be possibly due to other molecules or to a synergistic activity inside the phytocomplex.

To complete the preliminary evaluation of the cytotoxic effect of the extracts and pure molecules, their effect was tested against a human keratinocyte cell line (HaCat). None of the extracts or pure molecule showed cytotoxicity against the HaCat cell line, giving indication of safe use of the extracts.

### 2.5. Mutagenic Assay (Ames Test)

To better define the awareness on the efficacy and the safe possible use of *O. campechianum* extracts and EO, Ames test was performed.

All tested concentrations showed a number of treated/control colonies ratio less than 2, both in presence and in absence of S9 mix microsomal fraction (Table 6). Therefore, none of the texted extracts were considered potentially mutagenic. As regards the EO, a cytotoxic effect is noted at the highest concentrations. Our results pointed out the safety of *O. campechianum* extracts and EO with regard to genotoxicity [11].

## 3. Materials and Methods

### 3.1. Plant Material

The aerial parts of *O. campechianum* Mill. (stems, leaves and inflorescences; 20 Kg) were collected in January 2019, at early flowering stage at the CIPCA (Centre for Research, Postgraduate and Conservation of the Amazon, Santa Clara, Ecuador) of the Universidad Estatal Amazónica (UEA) (01°14′13″ S, 077°53′25″ W, 570 m), from a wild population in the Amazonian region of Napo (Arosemena Tola canton), Ecuador. The authentication was performed at the Amazon State University (UEA), using as reference a specimen previously deposited in the Herbarium ECUAMZ (voucher specimen: Radice 18070D). 

### 3.2. Preparation of Extracts

The EO was obtained by hydrodistillation of fresh aerial parts (5 Kg with 10 L of water) for 3 h in a stainless-steel distiller equipped with a Clevenger apparatus, performing three distinct distillations. The EO content was calculated on a moisture-free basis as average. The EO was dried over anhydrous sodium sulfate and stored in sealed amber vials at 4 °C.

Before proceeding with the alcoholic extraction, the dried aerial parts of *O. campechianum* (350 g) were milled through a 2 mm sieving ring of a Variable Speed Rotor Mill (Fritsch, Idar- Oberstein, Germany). Afterwards, 50 g of powdered crude drug was added to 500 mL of methanol (Sigma-Aldrich, Milano, Italy), while another 50 g to 350 mL of absolute ethanol (Sigma-Aldrich, Milano, Italy) and 150 mL of distilled water to obtain a 70% ethanolic solution. The extraction was performed by means of an ultrasonic bath (Branson Bransonic CPXH Digital Bath 3800F, Emerson, St. Louis, MO, USA) for 30 min at a temperature of 30 °C. The extracts were filtered and the methanolic solution were evaporated to dryness with a rotary evaporator (RV 10 digital, IKA^®^-Werke GmbH & CO. KG, Staufen im Breisgau, Germany), while the ethanolic one was evaporated to dryness with a rotary evaporator and then lyophilized to eliminate residual water. All the extractions were performed in triplicate and dried extracts stored at −20 °C. The EO content was calculated on a moisture-free basis as average.

### 3.3. Gas Chromatography Coupled to Mass Spectrometric and Flame Ionization Analyses 

The analysis method and composition of EO was previously described by Scalvenzi et al. (2019) [10].

### 3.4. HPLC-DAD-MS Analysis 

The analyses were performed using a JASCO modular HPLC system (Tokyo, Japan, model PU 2089) coupled to a diode array apparatus (MD 2010 Plus) and a FinniganMAT LCQ (ThermoQuest Corp./FinniganMAT; San Jose, CA, USA) mass spectrometer module linked to an injection valve with a 20 µL sampler loop. The column used was a Kinetex-C18 (15 × 0.46 cm, i.d., 5 µm, 100 Å, Phenomenex) at a flow rate of 0.7 mL/min and at a temperature of 30 °C. The mobile phase consisted of binary solvent system of water/formic acid 99/1 (solvent A) and methanol (solvent B). The gradient system adopted was: starting point at 95:5 *v*/*v* (A/B), gradual changing to 70:30 *v*/*v* in 50 min, isocratic condition up to 65 min, back to starting point at 70 min, re-equilibration of the system up to 85 min. Injection volume was 40 µL. The chromatograms were observed at 355 nm [24]. 

The mass experiments were performed on a FinniganMAT LCQ (ThermoQuest Corp./FinniganMAT; San Jose, CA, USA) mass spectrometer module, equipped with an ion trap mass analyzer and an ESI ion source electrospray, in negative ion mode. For ESI-MS and MS^2^ experiments, the parameters were set as follows; the capillary voltage was 3.5 kV, the nebulizer (N_2_) pressure was 20 psi, the capillary temperature was 300 °C, the auxiliary gas (N_2_) flow was 9 L/min, and the skimmer voltage was 40 V. The mass spectrometer was operated in the negative ion mode in the m/z range 100–1500. Standard commercial molecules of quercetin 3-*O*-β-rutinoside (rutin), caftaric, chlorogenic and rosmarinic acids were used to confirm the experimental data and to quantify these molecules in the extracts through the construction of calibration curves with solutions of concentrations from 1 to 50 μg/mL for rutin, 1 to 50 μg/mL for caftaric acid, 1 to 50 μg/mL for chlorogenic acid, and 10–350 μg/mL for rosmarinic acid. All standards were purchased by Extrasynthese (Cedex, France). The two dried extracts of *O. campechianum* were, respectively, solubilized in hydroalcoholic solution (ethanol 70%) at concentration of 4.0 mg/mL and in methanol at concentration of 2.5 mg/mL. 

### 3.5. DPPH Scavenging Activity

The DPPH assay was performed following the method by Cheng et al. (2006) [39]: briefly, the DPPH solution was placed on a 96-well plate containing different concentration of extract or pure compounds for 30 min in the dark at room temperature, then the microplates were analyzed with a microplate reader (680XR, Bio Rad, Laboratories, Inc., Hercules, CA, USA) and the absorbance was read in triplicate against a blank at 515 nm. The DPPH inhibition in percent was determined by the following formula: IDPPH% = [1 − (A1/A2)] × 100, where A1 was the DPPH absorbance with the extracts and A2 without extracts. Eight different concentrations (range: 20–0.16 μg/mL) of Trolox were prepared and used as positive control. The activity of the extracts was expressed as IC_50_, concentration providing 50% inhibition of the radical. All experiments were performed in triplicate.

### 3.6. ABTS Scavenging Activity

The ABTS scavenging activity was evaluated using the method of Horszwald and Andlauer (2011) [40]. EO, alcoholic extracts and pure molecules (rosmarinic acid, eugenol, and Trolox) were tested in a range of concentrations, respectively, 37.00–0.58 μg/mL, 133.33–2.08 μg/mL, and 13.33–0.21 μg/mL. Aqueous solution (7 mmol/L) of ABTS (10 mL) and 51.4 mmol/L aqueous solution of K_2_S_2_O_4_ (0.5 mL) were mixed to obtain a radical cation solution that has been adjusted spectrophotometrically to 0.7 ± 0.05 at 734 nm. After 6 min of incubation in the dark at room temperature, microplates were analyzed with a microplate reader (680XR, Bio Rad, Laboratories, Inc., Hercules, CA, USA), and the absorbance was read at 734 nm in triplicate and against a blank. Antioxidant activity of the samples was expressed as IC_50_, the concentration providing 50% radical inhibition. All experiments were assessed in triplicate and values were reported as mean ± standard deviation.

### 3.7. Antibacterial Activity 

The antibacterial activity has been evaluated against phytopathogenic and human pathogenic bacteria to preliminarily verify phytoiatric and health properties. *Pseudomonas syringae* pv. *syringae* ATCC 19310 and *Staphylococcus aureus* ATCC 6538 were used to determine MIC (Minimum Inhibitory Concentration) through microdilution method using 96-well microtiter plates [41]. Bacterial cultures were incubated overnight at 26 °C and 37 °C, respectively, in Tryptic Soy Broth (OXOID Ltd., Hampshire, UK). One hundred μL of sterile medium were used together with 100 μL of sample to perform serial dilutions of extracts, EO, rosmarinic acid and eugenol, previously dissolved in ethanol (50 mg/mL of stock solution), into all micro-wells.

One-hundred microliters of bacterial culture standardized to 2 × 10^7^ CFU/mL was added to the wells and incubated at 37 °C for 6 h and at 26 °C for 24 h, for human and phytopathogens, respectively. After the incubation period, 40 μL of water solution (20 mg/mL) of 2,3,5-triphenyl-tetrazolium chloride (Sigma-Aldrich, St. Louis, MO, USA) was added to each well and then incubated for 30 min: microbial growth was evaluated by microplate reader (680XR, Bio Rad, Laboratories, Inc., Hercules, CA, USA) at 415 nm. Chloramphenicol (concentration range 20–0.32 μg/mL) was used as a positive control. All determinations were made in triplicate.

### 3.8. Antimicrobial Activity against Candida spp. and Synergy Test

Antimicrobial and synergy tests were performed against three *Candida* spp. strains: *Candida albicans* (AIDS6; fluconazole resistant clinical strain isolated from a HIV patient), *Candida glabrata* (FLU 43976; resistant to fluconazole), and *Candida albicans* (ATCC 24433; from American Type Culture Collections; sensitive to fluconazole). The Minimum Inhibitory Concentration (MIC) and Minimum Fungicidal Concentration (MFC) of *O. campechianum* EO and fluconazole (positive control) were determined through microdilution method using 96-well microtiter plates [41], properly modified according our previous research [11]. Synergistic evidence of anti-*Candida* activity was checked through checkerboard test using EO and fluconazole at the same experimental conditions employed for MIC and MFC determination, as previously described. The synergistic activity was expressed as fractional inhibitory concentration (FIC_index_) and was calculated as follows,
FIC_index_ = FIC_EO_ + FIC_fluconazole_
where FIC_EO_ = MIC_EO+fluconazole_/MIC_EO_; FIC_fluconazole_ = MIC_fluconazole+EO_/MIC_fluconazole_. FIC_index_ values ≤ 0.5 mean the presence of synergistic effect, 0.5 < values ≤ 2 mean additive or indifferent effect, values > 2 mean antagonistic effect [36].

#### 3.8.1. Cell Lines and Culture Conditions

Adenocarcinomic human alveolar basal epithelial cells (A549) and human keratinocytes (HaCat) were purchased by Istituto Zooprofilattico Sperimentale della Lombardia e dell’Emilia-Romagna, Brescia, Italy, and maintained, respectively, in Ham’s F12 medium and DMEM containing 4.5 g/L and 1 g/L glucose. The cell lines were grown in 75 cm^2^ flasks and cultured in medium supplemented with 10% fetal bovine serum (FBS), 100 U/mL penicillin/streptomycin, and 2 mM L-glutamine in a humidified 5% CO_2_-95% air atmosphere at 37 °C until 80% confluence.

#### 3.8.2. Cell Viability Assay

Cell viability was determined by MTT colorimetric assay [24] as reflected by the activity of succinate dehydrogenase. Briefly, cells were seeded at the density of 2 × 10^4^ cells/well on a 96-well plate. After 24 h, cells were exposed to different concentrations of *O. campechianum* EO (1–500 µg/mL), *O. campechianum* 70% ethanolic extract (10–200 µg/mL), *O. campechianum* methanolic extract (10–200 µg/mL), eugenol and rosmarinic acid (1–100 µM) in a final volume of 200 µL of culture medium. Control culture was exposed to only vehicle (medium containing 2% FBS). After 24 h of incubation, 20 µL of MTT (5 mg/mL in phosphate-buffered saline, PBS) was added in each well and the plates were incubated for 4 h at 37 °C. The medium was removed and replaced with 100 µL DMSO to dissolve the formazan crystals. The extent of MTT reduction was measured spectrophotometrically at 570 nm using a microplate reader (680XR, Bio Rad, Laboratories, Inc., Hercules, CA, USA).

### 3.9. Mutagenic Assay

Mutagenicity assay was performed following the plate incorporation method with the histidine-requiring *Salmonella typhimurium* mutant TA98 and TA100 strains purchased by Molecular Toxicology Inc. (Boone, NC, USA; moltox.com). All strains (100 µL per plate of fresh overnight cultures) were checked with and without the addition of 0.5 mL of a 5% S9 exogenous metabolic activator (S9 mix). The lyophilized post-mitochondrial supernatant S9 mix (Aroclor 1254-induced, Sprague–Dawley male rat liver in 0.154 M KCl solution), commonly used for the activation of pro-mutagens to mutagenic metabolites (Molecular Toxicology, Inc., Boone, NC, USA) was stored at −80 °C before use. The concentration tested for all the samples were 5, 10, 20, 50, and 100 µL/plate of a stock solution 50 mg/mL. An amount of 0.5 mL of phosphate buffer or S9 mix for assays with metabolic activation was added to 2 mL molten top agar (0.6% agar, 0.6% NaCl, 0.5 mM L-histidine/biotin solution) at 46 °C, together with 0.1 mL of each sample solution at different concentrations, and 0.1 mL of fully-grown culture of the appropriate tester strain. The ingredients were thoroughly mixed and poured onto minimal glucose agar plates (1.5% agar in 2% Vogel–Bonner medium E with 5% glucose solution). DMSO was used as a negative control (100 µL/plate). Positive controls were prepared as follows: 2-aminoanthracene (2 µg/plate) for both strains with metabolic activation a 2-nitrofluorene (2 µg/plate) and sodium azide (2 µg/plate) for TA98 and TA100 without metabolic activator, respectively. The plates were incubated at 37 °C for 72 h and then the his+ revertants were checked and counted using a Colony Counter 560 Suntex (Antibioticos, Italy). A sample was considered mutagenic when the observed number of colonies was at least twofold over the spontaneous level of revertants [11]. All determinations were made in triplicate.

### 3.10. Statistical Analysis

Data are reported as mean ± standard error of the mean, and “n” was the number of independent experiments performed in triplicate. The statistical analysis for cell viability was calculated using one-way analysis of variance (ANOVA), followed by Dunnett’s Test. The results were considered significant with *p* < 0.01 compared to untreated cells.

## 4. Conclusions

The present work provides for the first time the chemical quantification of secondary metabolites in alcoholic extracts of *O. campechianum*, enhancing the knowledge on the phytochemicals produced by this species, currently known only for its EO.

On the basis of the in vitro results of this research, the crude drug and alcoholic extracts of *O. campechianum* could be proposed as safe alternative to *O. basilicum*, in consideration of the content of rosmarinic acid. On the other hand regarding the antimicrobial effect of *O. campechianum* EO, it could be suggested as a succedaneum of clove oil with a projection in human health and phytoiatric uses. 

In the final analysis, this species represents an interesting sustainable resource for future development of local economy and for trade at international level.

## Figures and Tables

**Figure 1 molecules-26-00084-f001:**
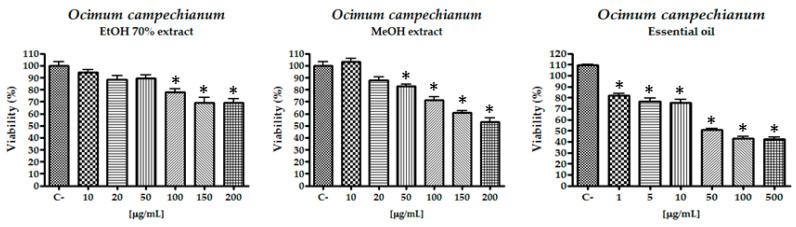
Cell viability of *O. campechianum* extracts and EO against A549 cell line. * *p* < 0.01

**Table 1 molecules-26-00084-t001:** Essential oil composition of *O. campechianum.*

No.	Component ^1^	Area% ^2^	AI exp ^3^	AI lit ^4^
1	α-pinene	0.3	929	932
2	camphene	0.1	944	946
3	sabinene	0.8	967	969
4	β-pinene	1.2	973	974
5	myrcene	2.3	987	988
6	**1,8-cineole**	**4.4**	1028	1026
7	*cis*-β-ocimene	1.3	1030	1032
8	*trans*-ocimene	0.4	1046	1044
9	linalool	1.6	1101	1095
11	*allo*-ocimene	1.7	1126	1128
12	mentha-1,5 dien-8 ol	0.3	1170	1166
13	α-terpineol	0.4	1189	1186
14	neral	0.1	1238	1235
15	**δ-elemene**	**4.2**	1338	1335
16	**eugenol**	**43.6**	1359	1356
17	α-copaene	0.5	137	1374
19	**β-elemene**	**8.1**	1388	1389
20	**β-caryophyllene**	**10.8**	1410	1416
21	γ-elemene	0.3	1427	1434
22	*trans*-α-bergamotene	0.6	1431	1435
23	α-caryophyllene	1.9	1451	1452
24	*allo*-aromadendrene	1.4	1455	1458
25	germacrene D	0.4	1477	1484
26	β-selinene	1.1	1484	1489
27	viridiflorene	0.4	1489	1496
27	**bicyclogermacrene**	**2.9**	1490	1500
28	germacrene A	1.0	1500	1508
29	germacrene B	1.6	1556	1559
30	spathulenol	1.5	1576	1577
31	caryophyllene oxide	1.9	1581	1582
	Total identified	97.1		

^1^ Components are listed in order of elution and their nomenclature is in accordance of the NIST (National Institute of Standards and Technology) library. ^2^ Relative peak areas, calculated by GC-FID. ^3^ AI exp: linear retention indices calculated on Varian VF-5 ms column. ^4^ AI lit: linear retention indices [21]. The main compounds were in bold

**Table 2 molecules-26-00084-t002:** Main constituents identified in extracts of *O. campechianum* by HPLC-DAD, high-performance liquid chromatography/electrospray ionization mass spectrometry (HPLC-ESI-MS), and MS^2.^

Compound ^a^	70% EtOH Extract mg/g d.e. (mg/g drug) ^b^	MeOH Extract mg/g d.e. (mg/g Drug) ^b^	UV ʎ_max_ (nm)	[M − H]^−^ (*m*/*z*)	MS^2^ (*m*/*z*) Base Peak
Caftaric acid	0.35 ± 0.01 (0.07 ± 0.00)	0.57 ± 0.01 (0.03 ± 0.00)	327	311	149
Chlorogenic acid	0.27 ± 0.01 (0.05 ± 0.00)	0.05 ± 0.01 (0.003 ± 0.000)	327	353	191
Rutin	0.63 ± 0.01 (0.12 ± 0.01)	0.56 ± 0.01 (0.02 ± 0.00)	255, 355	609	301
Rosmarinic acid	22.3 ± 0.2 (4.33 ± 0.01)	21.8 ± 0.3 (1.17 ± 0.02)	327	359	161

^a^ Compounds are listed in order of elution, ^b^ d.e. = dried extract.

**Table 3 molecules-26-00084-t003:** DPPH and ABTS IC_50_ (µg/mL) of *O. campechianum* extracts, EO, and main compounds.

Extracts and Compounds	DPPH IC_50_	ABTS IC_50_
*O. campechianum* 70% EtOH extract	11.10 ± 1.13	8.58 ± 0.53
*O. campechianum* MeOH extract	52.15 ± 2.72	15.15 ± 0.52
Rosmarinic acid	4.31 ± 0.34	1.93 ± 0.07
Trolox (positive control)	3.66 ± 0.29	2.14 ± 0.25
*O. campechianum* EO	7.77 ± 0.07	3.18 ± 0.29
Eugenol	5.64 ± 0.27	8.58 ± 0.53

**Table 4 molecules-26-00084-t004:** MICs of *O. campechianum* extracts, EO, and main compounds.

Extracts and Compounds	MIC (μg/mL) *Staphylococcus aureus* (ATCC 6538)	MIC (μg/mL) *Pseudomonas syringae* pv. *syringae* (ATCC 19310)
*O. campechianum* 70% EtOH extract	>2000	>2000
*O. campechianum* MeOH extract	>2000	>2000
Rosmarinic acid	>2000	>2000
Cloramphenicol (positive control)	10	2.5
*O. campechianum* EO	>2000	250
Eugenol	1000	500

**Table 5 molecules-26-00084-t005:** Synergistic effect of *O. campechianum* EO employing fluconazole as synthetic active drug against *Candida* spp.

	FIC_index_ MIC (EO + FLU)	FIC_index_ MFC (EO + FLU)	EO MIC MFC		FLU MIC MFC	
*C. albicans* (AIDS6)	1.031 (1563 − 4) *	1.031 (1563 − 4) *	1563	1563	128	128
*C. glabrata* (FLU 43976)	0.562 (781 − 8) *	0.562 (781 − 8) *	1563	1563	128	128
*C. albicans* (ATCC 24433)	1.004 (3135 − 0.5) *	1.004 (3135 − 0.5) *	3135	3135	128	128

FIC_index_: fractional inhibitory concentration index; FIC_index_ ≤ 0.5 means the presence of synergistic effect; 0.5 ≤ FIC_index_ ≤ 2 means additive or indifferent effect; FIC_index_ >2 means antagonistic effect [11,36].* Concentration in µg/mL of EO and FLU, respectively, corresponding to FIC_index_. MIC: Minimum Inhibitory Concentration (µg/mL); MFC: Minimum Fungicidal Concentration (µg/mL). EO: *O. campechianum* EO; FLU: fluconazole.

**Table 6 molecules-26-00084-t006:** Treated/control (t/c) ratio of colony count in the Ames test.

Extracts	*O. Campechianum* EO					*O. Campechianum* 70%EtOH Extract				*O. Campechianum* MeOH Extract			
	TA98 (t/c) *		TA100 (t/c) *			TA98 (t/c)*		TA100 (t/c) *		TA98 (t/c) *		TA100 (t/c) *	
Conc.% (mg/plate)	−S9	+S9	−S9		+S9	−S9	+S9	−S9	+S9	−S9	+S9	−S9	+S9
**5 (0.25 mg/plate)**	0.9	0.7	1.0	0.8		0.7	1.0	0.9	0.8	0.4	0.9	1.2	0.9
**10 (0.5 mg/plate)**	0.9	0.5	1.0	0.8		0.7	1.2	0.9	1.0	0.7	0.9	1.1	0.9
**20 (1 mg/plate)**	0.9	0.6	0.9	0.7		0.7	1.0	0.9	0.9	0.5	0.7	0.9	0.9
**50 (2.5 mg/plate)**	1.1	0.3	0.5	0.2		1.0	0.5	1.1	1.0	0.3	0.8	1.0	0.9
**100 (5 mg/plate)**	0.3	0.0	0.0	0.2		0.7	0.7	1.0	0.7	0.6	0.5	1.1	1.0
**C+ ****	4.4	4.3	5.4	4.6		4.4	4.3	5.4	4.6	3.8	4.3	5.6	4.6

* t/c means the ratio between the number of colonies of *Salmonella* strains grown in presence of EO and those of the negative control (DMSO). ** C+: positive controls, that were: 2-aminoanthracene (2 µg/plate) and 2-nitrofluorene (2 µg/plate) for TA98 with and without metabolic activator (S9 mix) respectively; 2-aminoanthracene (2 µg/plate) and sodium azide (2 µg/plate) for TA100, with and without metabolic activator (S9 mix) respectively.

## Data Availability

Data is contained within the article.

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
