# Peer review of "Ocimum campechianum Mill. from Amazonian Ecuador: Chemical Composition and Biological Activities of Extracts and Their Main Constituents (Eugenol and Rosmarinic Acid)"

_molecules, 2020, doi:10.3390/molecules26010084_

Round 1

Reviewer 1 Report

In the present study the others study the chemicals composition composition and somes biological activities of essential oils and hydroalcoholic extracts and main constituents of the aerial parts of Ocimum campechianum Mill. from Amazonian Ecuador. This is a very interesting paper, the other hand this work presents some minor corrections according to the following comments.

Comments to Authors:

1- Page 6, line 211:

  • The others used the aerial parts of Ocimum campechianum, Please precise? (Flowers are part of it).
  • The aerial parts of campechianum Mill. (20 Kg) were collected in 2016, (The season and the month of collection must be specified).

2- For the antibacterial Activity, why the others did not determine the inhibition zone diameter (IZ) and the minimum bactericidal concentration (MBC)?

3- Lines 162; 222; 224; 307; 308; 311; 318; 319; 326 and in all the text standardize the writing of the unit ml and µl.

4- Page 5, Figure 1: O.campechianum EtOH 70% (correct the title of the x-axis).

5- Page 5: Enter figure 1 in the corresponding text.

6- Page 8, line 306: “cells were seeded at the density of 2 x 104” correct the number of cells.

Author Response

In the present study the others study the chemicals composition and some biological activities of essential oils and hydroalcoholic extracts and main constituents of the aerial parts of Ocimum campechianum Mill. from Amazonian Ecuador. This is a very interesting paper, the other hand this work presents some minor corrections according to the following comments.

Authors: The authors are grateful for the acknowledgement. The manuscript has been revised according to the suggestions, and responses to the reviewer have been inserted after each reviewer's comment.

Comments to Authors:

1- Page 6, line 211:

- The others used the aerial parts of Ocimum campechianum, Please precise? (Flowers are part of it).

Authors: For the preparation of the extracts all the apogee parts of the plant (stems, leaves and inflorescences) have been used. The information has been added to the revised manuscript (new line 283).

- The aerial parts of O. campechianum Mill. (20 Kg) were collected in 2016, (The season and the month of collection must be specified).

Authors: The information has been added. We apologize, in the original manuscript there was a typing error: the year is 2019, the month is January. (line 284).

2- For the antibacterial Activity, why the others did not determine the inhibition zone diameter (IZ) and the minimum bactericidal concentration (MBC)?

Authors: The agar-disk diffusion method is not an appropriate method to evaluate the antimicrobial activities of essential oil: potential strong inhibitors having low water solubility gave a poor or even negative result in the agar diffusion test. It is therefore wrong to conclude that an essential oil without resulting in an inhibition zone in the agar diffusion test is without any antimicrobially active constituents; or in other words, antimicrobially active compounds are easily overlooked by this method (BaÅŸer, K. H. C., Buchbauer, G. Handbook of essential oils: science, technology and applications, 2nd ed.; CRC Press/Taylor & Francis, Boca Raton, USA 2015, 442). The use of microdilution method is a better appropriate approach when one of the extracts is an essential oil.

3- Lines 162; 222; 224; 307; 308; 311; 318; 319; 326 and in all the text standardize the writing of the unit ml and µl.

Authors: The units of measurement of the text have been aligned as requested.

4- Page 5, Figure 1: O. campechianum EtOH 70% (correct the title of the x-axis).

Authors: Figure 1 has been modified in accordance with the reviewer's commentary.

5- Page 5: Enter figure 1 in the corresponding text.

Authors: The reference to Figure 1 has been inserted in the text (line 224).

6- Page 8, line 306: “cells were seeded at the density of 2 x 104” correct the number of cells.

Authors: the text has been modified in accordance with the reviewer's commentary. (new line 398)

Reviewer 2 Report

The work presented for the review is interesting, although taking into account the character of the journal, it requires additional chemical analyzes of plant material used for the experiments. The description of the obtained results is sometimes unclear (even misleading), e.g. the Authors did not investigate the chemical composition of the essential oil (EO), as suggested in the first sentence of the abstract and in the aim of work (lines 19-21; chemical characterization of the EO was not carried out). Taking into account the presented results of biological activity of EO, this analysis is necessary here (e.g. GC-MS) to discuss the results and to draw reliable conclusions. This is not acceptable to present such an important part of the experiment in the other article (already published – Chapter 3.3).

Moreover, the reliable results regarding the antioxidant activity of plant raw materials may be obtained only by applying 2-3 different analytical methods. One-test analysis may be questionable.

Minor comments

Any abbreviations should be explained when first used.

The manuscript should be revised in terms of quality of description, which is sometimes chaotic. This is especially visible in the Abstract.

What are “main constituents” or “main molecules” mentioned trough the manuscript? It would be better to name these constituents.

Materials: What are “aerial parts” and what was the developmental stage of the plant during the collection (vegetative stage; flowering or maybe fruiting?).

Line 220: The yield of essential oil? This parameter is estimated when regards plantations of medicinal plants. In the situation described in this work this is rater content. The results (EO content) should be presented in the next chapter (Results and Discussion).

Strong sides

The introduction chapter clearly presents the issues regarding the need to both: use and protect wild-growing medicinal and aromatic plants. The results concerning antibacterial activity are very interesting.

Author Response

The work presented for the review is interesting, although taking into account the character of the journal, it requires additional chemical analyzes of plant material used for the experiments. The description of the obtained results is sometimes unclear (even misleading), e.g. the Authors did not investigate the chemical composition of the essential oil (EO), as suggested in the first sentence of the abstract and in the aim of work (lines 19-21; chemical characterization of the EO was not carried out). Taking into account the presented results of biological activity of EO, this analysis is necessary here (e.g. GC-MS) to discuss the results and to draw reliable conclusions. This is not acceptable to present such an important part of the experiment in the other article (already published – Chapter 3.3).

Authors: the authors are grateful for the recommendations. The manuscript has been revised according to the suggestions, and responses to the reviewer have been inserted after each reviewer's comment.

Moreover, the reliable results regarding the antioxidant activity of plant raw materials may be obtained only by applying 2-3 different analytical methods. One-test analysis may be questionable.

Authors: Authors agree with reviewer and have added another analysis of antioxidant activity (ABTS) to confirm DPPH test results.

Minor comments

Any abbreviations should be explained when first used.

The manuscript should be revised in terms of quality of description, which is sometimes chaotic. This is especially visible in the Abstract.

Authors: The abstract and the rest of the manuscript have been extensively revised from a linguistic point of view.

What are “main constituents” or “main molecules” mentioned trough the manuscript? It would be better to name these constituents.

Authors: The authors have been aligned the revised manuscript using “these constituents”.

Materials: What are “aerial parts” and what was the developmental stage of the plant during the collection (vegetative stage; flowering or maybe fruiting?).

Authors: the apogee parts of the plant (stems, leaves and inflorescences) have been used. The developmental stage is early flowering stage. The informations have been added to the revised manuscript (new line 283).

Line 220: The yield of essential oil? This parameter is estimated when regards plantations of medicinal plants. In the situation described in this work this is rater content. The results (EO content) should be presented in the next chapter (Results and Discussion).

Authors: Authors agree with reviewer and change “EO yield” with “EO content”. They delete the result of EO content (new line 294) and reported it in Results and discussion.

Strong sides

The introduction chapter clearly presents the issues regarding the need to both: use and protect wild-growing medicinal and aromatic plants. The results concerning antibacterial activity are very interesting.

Reviewer 3 Report

The manuscript by Tacchini et al. (Molecules-975354) deals with the radical scavenger activity and the antimicrobial activity against Pseuidomonas syringae pv. syringae and Candida of essential oils and hydroalcoholic extracts obtained from aerial parts of Ocimum campechianum.

In my opinion the study is interesting and well conceived. The experimental methods and results are clearly described. The discussion takes into due consideration the literature data available on the subject matter.

I recommend, therefore, to publish the manuscript in its present form.

Author Response

The manuscript by Tacchini et al. (Molecules-975354) deals with the radical scavenger activity and the antimicrobial activity against Pseuidomonas syringae pv. syringae and Candida of essential oils and hydroalcoholic extracts obtained from aerial parts of Ocimum campechianum.

In my opinion the study is interesting and well conceived. The experimental methods and results are clearly described. The discussion takes into due consideration the literature data available on the subject matter.

I recommend, therefore, to publish the manuscript in its present form.

Authors: The authors are grateful for the acknowledgement.

Reviewer 4 Report

Dear Authors,

Firstly I would like to congratulate you on an attempt to discuss such an important topic and to present the findings however, I have several concerns and I have provided you with the number of comments. I have attempted to divide them in Major and Minor comments for your pursuit. I sincerely hope that this comments/suggestions assist in the improvements of the manuscript.

Major:

  1. Manuscript will require editing in English as there are several academic and grammatical errors (as an example, line 42 – 45 is all one sentence with word species used to many times. It will bring a better context to the whole article which is rather interesting).
  2. The title is very confusing as I cannot distinguish what it compared to what? Please clarify the title.
  3. Introduction is rather brief in particular for such a diverse and important topic. There is no adequate information on the previous studies in this area and after a quick search of literature, there are several hundred articles available. Please include adequate literature background that relates to the other Amazonian species of interest before discussing the current one.
  4. Line 71 – 80 is very confusing. Is this composition already published elsewhere or these are the findings of your study? If it is the previous publication then why is this listed? If it is current findings, again, why is this listed and what were the methods used?
  5. Line 227 – how were the extracts evaporated to dryness? Was there a stream of N2 being used or was this air-dried? More information is needed.
  6. It is very common to include two antioxidant tests such as DPPH and one of the ABTS/CUPRAC/FRAP or others as it is accepted that only one total antioxidant inhibition test is insufficient to present the adequate results. Please elaborate.
  7. Suggesting O. campechianum as an alternative drug to O basilicum? Why is this? How can this be based on the provided results?
  8. Conclusion are not supported by the observations. The information provided is only in its primary stage (true) but extrapolating this information for the use in humans is still a considerable way away from reality. There are needs to establish toxicological and sensitivity data initially, animal models, etc… the findings form in vitro studies cannot be that easily translated into the human health benefits. Please reword the conclusion.

Minor:

  1. The information about the cooperation between the universities (lines 45-49 are unnecessary. Please remove from the introduction.
  2. Sentence in Line 64-65 is unnecessary… please delete (In fact, the combination…)
  3. Line 222 – do not start the sentence with a number.
  4. Please review the references and referencing style to be consistent for the journal requirements.

Author Response

Dear Authors,

Firstly I would like to congratulate you on an attempt to discuss such an important topic and to present the findings however, I have several concerns and I have provided you with the number of comments. I have attempted to divide them in Major and Minor comments for your pursuit. I sincerely hope that this comments/suggestions assist in the improvements of the manuscript.

 Authors: the authors are grateful for the recommendations. The manuscript has been revised according to the suggestions, and responses to the reviewer have been inserted after each reviewer's comment.

Major:

  • Manuscript will require editing in English as there are several academic and grammatical errors (as an example, line 42 – 45 is all one sentence with word species used to many times. It will bring a better context to the whole article which is rather interesting).

Authors: The authors extensively revised from a linguistic point of view.

  • The title is very confusing as I cannot distinguish what it compared to what? Please clarify the title.

Authors: The authors revised the title, taking also to account the recommendations of other reviewers.

  • Introduction is rather brief in particular for such a diverse and important topic. There is no adequate information on the previous studies in this area and after a quick search of literature, there are several hundred articles available. Please include adequate literature background that relates to the other Amazonian species of interest before discussing the current one.

Authors: The authors extensively revised the introduction, inserting considerations related to the other Amazonian species of interest in this area.

  • Line 71 – 80 is very confusing. Is this composition already published elsewhere or these are the findings of your study? If it is the previous publication, then why is this listed? If it is current findings, again, why is this listed and what were the methods used?

Authors: The authors better clarified in the text the composition of essential oil and inserted a table that specified the findings of their study (new lines 95-112).

  • Line 227 – how were the extracts evaporated to dryness? Was there a stream of N2 being used or was this air-dried? More information is needed.

Authors: The information has been added (new lines 303-306)

  • It is very common to include two antioxidant tests such as DPPH and one of the ABTS/CUPRAC/FRAP or others as it is accepted that only one total antioxidant inhibition test is insufficient to present the adequate results. Please elaborate.

Authors: Authors agree with reviewer and have added another analysis of antioxidant activity (ABTS) to confirm DPPH test results. They also better discussed their findings.

  • Suggesting O. campechianum as an alternative drug to O basilicum? Why is this? How can this be based on the provided results?

Authors: The authors better clarified the sentence. O. campechianum could be suggested as an alternative for O. basilicum, with reference to the content of rosmarinic acid.

  • Conclusions are not supported by the observations. The information provided is only in its primary stage (true) but extrapolating this information for the use in humans is still a considerable way away from reality. There are needs to establish toxicological and sensitivity data initially, animal models, etc… the findings form in vitro studies cannot be that easily translated into the human health benefits. Please reword the conclusion.

 Authors: Authors reworded the conclusion

Minor:

  • The information about the cooperation between the universities (lines 45-49 are unnecessary. Please remove from the introduction.

Authors: The authors have better clarified in the revised introduction the importance of university cooperation with regard to the topics covered in this paragraph of the manuscript.

  • Sentence in Line 64-65 is unnecessary… please delete (In fact, the combination…)

Authors: Literature (Trends in Plant Sci 19(3):140–145, 2014;. Trends Food. Sci. Tech. 2007, 18(8), 434-444.) and International Agencies (EMA, FDA,WHO) emphasize the importance of a rigorous scientific approach to the study and standardization of medicinal plants. In this sense, the authors consider the sentence as relevant.

  • Line 222 – do not start the sentence with a number.

Authors: The manuscript has been modified according to the reviewer's indications

  • Please review the references and referencing style to be consistent for the journal requirements.

Authors: The references have been reviewed.

Round 2

Reviewer 4 Report

Dear Authors

I would like to congratulate you on addressing all of the comments successfully and substantially updating the manuscript.

My only comment is to proof read the article once more and adjust for some minor typographical and grammatical errors.

I have no further comments/suggestions. Looking forward to seeing this in the final version.

Best of luck with your research.